# Association between Hope for the Future and Academic Performance in Adolescents: Results from the K-CHILD Study

**DOI:** 10.3390/ijerph191911890

**Published:** 2022-09-20

**Authors:** Tomoka Kashiwabara, Takeo Fujiwara, Satomi Doi, Yui Yamaoka

**Affiliations:** 1Department of Global Health Promotion, Tokyo Medical and Dental University (TMDU), Tokyo 113-8510, Japan; 2Japan Society for the Promotion of Science, Tokyo 102-0083, Japan

**Keywords:** hope for the future, time usage, academic performance, resilience, adolescent

## Abstract

In Japan, having hope for the future is emphasized in school. This study aimed to examine the association between hope for the future and academic performance among Japanese adolescents. Data were taken from the population-based Kochi Child Health Impact of Living Difficulty (K-CHILD) study conducted in 2016. Participants included 3477 adolescents in the eighth grade (i.e., 13–14 years old) in Kochi Prefecture. Information on hope for the future, self-rated academic performance, and time used for studying or playing was provided by the adolescents via a questionnaire. The question on resilience was answered by their caregivers. Propensity-score matching was applied for the allocation of hope for the future. Overall, 2283 adolescents (65.6%) had some form of hope for the future. Adolescents having hope for the future showed a higher self-rated academic performance (β = 0.21, 95% confidence interval (Confidence Interval (CI) = 0.10 to 0.32)), spent more time studying except in class (Odds Ratio (OR) = 1.89, 95% CI = 1.37 to 2.61), read more books (OR = 1.45, 95% CI = 1.19 to 1.75), and had a higher score of resilience (β = 1.48, 95% CI = 0.98 to 1.98), while the time to watch TV or DVDs was not different (*p* = 0.61). Our results highlight the importance of encouraging adolescents to have hope for the future to promote academic performance.

## 1. Introduction

Academic performance among adolescents is highly related to their health status. Adolescents who have poor academic performance have a higher risk of negative lifetime outcomes, such as mental health problems [1,2], substance use [3], or delinquency [4]. Hence, it is important to show the modifiable determinants of academic performance, such as having hope for the future. The reason why we chose hope for the future among many modifiable determinants is that having hope for the future is emphasized in school, especially in Japan [5]. Children are often asked “What do you want to become in the future?” by teachers, parents, or relatives.

Hope for the future was used in a study of self-esteem and academic performance among early Japanese adolescents [5]. In the study, hope for the future was measured by four items: (a) I have already decided what I want to become in the future; (b) I have a thing that I try to do in the future; (c) I think my future life is bright; (d) I do not think about how my future will be (reverse item). The authors found that changing the process of hope for the future was not associated with academic performance [5]. However, as the previous study’s population had a small sample size (*n* = 755) and was not population-based, a further study that addresses the limitations is needed.

In other countries, the association between hope for the future and academic performance has not been revealed. As a similar concept, future orientation or planning is only used in some studies. For example, in China and Iran, a significant association between future orientation or planning and academic achievement was demonstrated [6,7]. Hence, the main purpose of this study is to investigate the association between hope for the future and academic performance. Future orientation or planning may be associated with higher academic performance because an academic score is needed to obtain the occupation that he or she desires. However, it is not clear whether future orientation is determined by himself or herself; that is, hope for the future can have a more internal motive basis, whereas future orientation or planning has a more instrumental motive basis. As previous psychological studies revealed, internal motives would be more important for achievement in later life [8].

Moreover, other behaviors that are proven to be associated with academic performance, such as the amount of time spent on doing their homework and schoolwork [9], and the amount of time spent on playing video games [10], need to be assessed. The previous study about the future time perspective shows that time management disposition has a significant positive association with future positiveness and future plans, which can be seen as a similar concept of hope for the future. The previous study also showed that time planning has a significant indirect effect on academic achievement through time efficacy [11]. We postulate that this association may be adaptable to hope for the future. However, to our knowledge, there is a dearth of literature examining hope for the future and time used for studying or playing in adolescents.

Resilience was also noted as a predictor of academic performance [12]. In addition, future orientation, which is a similar concept to hope for the future, has been shown to greatly impact the development of resilience among maltreated youth [13]. We hypothesize that this association may also be adaptable to hope for the future, but resilience has never been addressed as a mediator between hope for the future and academic achievement.

Hence, another purpose of this study is to investigate the association of hope for the future with academic performance, time usage, and resilience. However, it is of note that hope for the future can vary by demographics, socioeconomic status, and daily activities, which can be a confounder of these associations [14,15]. Propensity score matching (PSM) is a method of balancing covariates between the exposure and non-exposure groups using propensity scores in a real-life situation where a randomized experimental design is not possible. As multiple factors can be confounded simultaneously, propensity score matching will enable us to conduct a pseudo-randomized controlled trial, which might estimate the causal relationship better than a cross-sectional study. In addition, this method is more robust in handling endogeneity bias [16].

In this study, using propensity score matching, we expected to find a positive association between hope for the future and academic performance among Japanese adolescents. In addition, we also hypothesized that hope for the future is associated with adolescents’ time usage and resilience. Speaking of time usage, it was anticipated that adolescents having hope for the future would spend less time playing games, watching TV or DVDs, and spend more time reading books or studying except when in class.

## 2. Materials and Methods

### 2.1. Sampling Method

We used data from the Kochi Child Health Impact of Living Difficulty (K-CHILD) study, a population-based cross-sectional study conducted in Kochi Prefecture, Japan, in 2016. The K-CHILD study aimed to examine the living environment and health of children in Kochi Prefecture. Participants included first, fifth, eighth, and eleventh-grade students and their caregivers in all public, private, and special needs schools in the prefecture, except for correspondence course high schools and a special needs school. Further details are described elsewhere [17].

The current study targeted students in the second grade in junior high schools (i.e., 13–14 years old in the eighth grade) and their caregivers. The self-administered anonymous questionnaires for the children and their caregivers were distributed to a total of 6192 eighth graders, and 3634 questionnaires were returned in anonymous envelopes to the Office of Children and Family Services in Kochi City by mail or to the schools in person (response rate: 54.7%). Of the returned questionnaires, 3624 were valid except for those completely blank (*n* = 10). We also excluded the ones which did not report on hope for the future (*n* = 147). Finally, 3477 subjects were used as the analytical sample in this study (Figure 1).

### 2.2. Measurements

#### 2.2.1. Hope for the Future

Hope for the future was assessed using the question, “Have you already decided what you want to become in the future?” The children responded with either “1” (yes) or “0” (no). The same question was used to measure hope for the future in a previous study on Japanese adolescents [5].

#### 2.2.2. Self-Related Academic Performance

Self-rated academic performance was assessed using the question, “How do you rate your academic performance in a class?” The children responded with “1” (higher than average) to “5” (lower than average), or “6” (I do not know). In this study, scores were used in reverse order so that a higher score indicates a higher level of academic performance. When we analyzed the relationship between hope for the future and self-related academic performance, we excluded “6” (I do not know) from the analysis as it was done in a previous study [17].

#### 2.2.3. Time Usage

This survey asked about the time spent on four activities such as playing games, watching TV or DVDs, reading books, and studying outside of class. Children were asked, “How often do you play games in a week?”, “How often do you watch TV or DVD in a week?” and “How often do you read books in a week?” Their responses ranged from “more than 2 h in a day”, “1 to 2 h in a day”, “less than 1 h in a day”, “4 to 5 days in a week”, “2 to 3 days in a week”, and “1 day in a week”, to “almost never”. For the statistical analysis, this categorical variable was changed into dichotomous variables. As for the time playing games and watching TV or DVDs, following the American Academy of Pediatrics guideline on screen time for children [18], “More than 2 h in a day”, and “1 to 2 h in a day” were categorized as “1” and “less than 1 h in a day”, “4 to 5 days in a week”, “2 to 3 days in a week”, “1 day in a week”, and “almost never” were categorized as “0”. As for the time spent reading and studying, reading and studying outside of class are recommended in school in Japan [19]. “More than 2 h in a day”, “1 to 2 h in a day”, “4 to 5 days in a week”, “2 to 3 days in a week”, “1 day in a week”, and “less than 1 h in a day” were categorized as “1” and “almost never” was categorized as “0”.

#### 2.2.4. Resilience

Children’s resilience was assessed using the Children’s Resilient Coping Scale (CRCS) developed for use in the Japanese context regarding resilience and coping skills [11]. CRCS consists of eight items: (1) speaks positively about their future; (2) tries to do their best; (3) able to take teasing or mean comments well; (4) knows how to properly greet others; (5) able to get ready for school, study, and do his/her chores without directions; (6) seeks appropriate advice when necessary; (7) able to give up on things they want or do things that they do not like to do for better future outcomes; and (8) able to ask questions to learn about what they do not understand. Caregivers answered each item from “0” (never) to “4” (very frequently). This scale showed high internal consistency (Cronbach’s alpha = 0.80) and sufficient validity in the literature [20]. A higher score indicates a higher level of resilience (range: 0–32).

#### 2.2.5. Covariates

In the questionnaire for caregivers, the demographic variables, socio-economic status, and physical and mental health of caregivers were assessed. Demographic variables included child’s sex (male or female), child’s height (<150, 150–159.9, 160–169.9, ≥170 cm), child’s weight (<40, 40–49.9, 50–59.9, ≥60 kg), maternal and paternal age (<40, 40–49, ≥50 years old), maternal and paternal education level (high school or less, some college education, college education or higher, other/unknown), maternal and paternal occupation (full-time job, part-time job, self-employed or others, not working), marital status (married or unmarried), whether the child is living with maternal and paternal grandparents (yes or no), and whether the child is living with siblings (yes or no). Socioeconomic status was assessed by annual household income (<JPY 3,000,000 (≒USD 27,777 in 2016), JPY 3,000,000–5,999,999, ≥JPY 6,000,000, or unknown), whether the household is receiving public assistance (yes or no), whether they are experiencing economic difficulties (yes or no), and whether they have experienced lack of daily necessities (yes or no).

Caregivers also answered self-rated physical health (good, average, poor). The caregiver’s mental health was measured using the Japanese version of the Kessler 6 (K6) [21], which comprises six items on a scale of “0” (all of the time) to “4” (none of the time). The total scores range from 0 to 24 and were divided into three levels using a score of 4/5 and 12/13 as cut-off points. Higher scores indicate a higher risk of having psychological distress. Caregivers also answered maternal and paternal smoking habits with either of the three responses (“I currently smoke”, “I smoked in the past”, or “I have never smoked”).

Lastly, the child’s mental health was measured using the Japanese version of the Depression Self-Rating Scale (DSRS) [22] in the questionnaire for children. In this study, children answered 15 items on a scale of “0” (never) to “2” (most of the time). The total scores range from 0 to 30 and were divided into two levels using a score of 16 as a cut-off point [23].

#### 2.2.6. Statistical Analysis

First, univariate linear regression analyses (crude model) and multivariate linear regression analyses (adjusted model) were conducted to examine the association between hope for the future and academic performance and the child’s resilience score. The adjusted model included the covariates listed above, such as demographic variables, socioeconomic status, self-rated academic performance in class, physical and mental health of mothers and fathers, and mental health of children. The rationale for these covariates is as follows: According to previous studies on future orientation, characteristics of parents, such as socioeconomic status or health status of parents were known to have an impact on the future orientation of children [24,25,26,27,28,29]. In addition, as children were less likely to have future-related thoughts if they had experienced childhood adversity like parental separation and divorce [30], we included marital status as covariates. In addition, as an association between a supportive home environment and adolescent future orientation was reported [31,32], covariates of siblings or grandparents were included. Further, previous studies revealed a significant association of future orientation with the child’s sex [33,34] and eating behavior, which is related to height and weight [35]. Adolescents’ mental health is associated with and conceptually close to future orientation [36,37,38,39,40]. We also conducted the crude and multivariate logistic regression analysis for the time used for studying or playing. Further, multicollinearity was checked by the variance inflation factor (VIF) and confirmed all VIFs < 10.

Second, propensity-score (PS) matching was used to compare adolescents who have hope for the future with those without, which can remove bias due to all observed covariates and make the observational study pseudo-experimental [16]. We included the same covariates used in the multivariate model. Missing data were categorized into one group of each covariate. PS matching was performed using the command “psmatch2” with a logistic regression model in STATA. We conducted 1 to 1 optimal matching, with a caliper width equal to 0.01, and no replacement was applied for PS matching. Standardized bias was used to assess the balance of potential confounders within the matched pairs. Then, conditional regression analysis was performed to examine the association between hope for the future and academic performance, and between hope for the future and CRCS total score, using the matched pairs. Similarly, the analyses for time spent playing games, watching TV or DVD, reading books, and studying outside of class were conducted using conditional logistic regression analysis.

All analyses were performed using STATA 14.2.3. To address multiple comparisons in an outcome-wide study [41], as we have six outcomes (academic performance, time to play games, time to watch TV or DVDs, reading books, studying except class, and resilience), we considered alpha = 0.05/6 = 0.0083 as a significant level, applying the Bonferroni correction.

## 3. Results

### 3.1. Distribution of Characteristics

Table 1 describes the characteristics of adolescents with and without hope for the future before and after propensity score matching. Overall, 2283 adolescents (65.6%) had hope for the future while 1194 adolescents (34.3%) had none. Female students and those having lower depressive symptoms were more likely to have hope for the future before PS matching. After PS matching, the standard bias of almost all covariates was less than 5%, which means that the bias of covariates within matched pairs was significantly lower than that of unmatched samples.

Table 2 shows the distribution of time usage. About 70 percent of adolescents play games for less than 1 h per day. On the other hand, about 70 percent of them watch TV or DVDs for more than 1 h per day. The percentage of adolescents who almost never read books was approximately 30%, while adolescents who almost never study except in class was 10% or less.

### 3.2. Propensity Score Matching

Table 3 shows the results of academic performance and resilience by univariate linear regression analyses (crude model) and multivariate linear regression analyses (adjusted model) before and after PS matching. All crude models and adjusted models before and after PS matching showed similar statistical differences. In the model before PS matching, adolescents with hope for the future showed higher self-rated academic performance (β = 0.19, 95% confidence interval (CI = 0.10 to 0.28)) and a higher score of resilience (β = 1.51, 95% CI = 1.11 to 1.91). In the model after PS matching, the significant difference in both academic performance and resilience remained (both *p* < 0.001).

Table 4 describes the results of time usage by logistic regression analyses before and after PS matching. Having hope for the future was also associated with studying except in class (OR = 1.45, 95% CI = 1.19 to 1.75) and reading books (OR = 1.89, 95% CI = 1.37 to 2.61). This result did not change before and after PS matching. There was no association between having hope for the future and time spent watching TV or DVDs. The association between hope for the future and time playing games was marginal: those who have hope for the future were 19% less likely to play games for 1 h or more per day (OR = 0.81, 95% CI = 0.67 to 0.98), which was not statistically significant after the Bonferroni correction.

## 4. Discussion

This study aimed to describe the association between hope for the future and academic performance, time usage, such as studying time, reading time, gaming time, and time watching TV/DVDs, and resilience among adolescents in Japan.

The results of the current study indicate that hope for the future was significantly associated with subjective academic performance. Though the significant association between future orientation or planning and academic achievement was demonstrated in previous studies in China and Iran [6,7], the association between hope for the future and academic performance has not yet been revealed in other countries. In Japan, the previous study that examined the association between hope for the future and academic performance among early Japanese adolescents showed that changing the process of hope for the future was not associated with academic performance [5].

This study adds to the literature in China and Iran by not showing future orientation but that hope for the future, which is based on an internal motive [8] and can be inspired in school, was associated with academic performance among Japanese adolescents. In addition, this study employed propensity score matching to eliminate measured bias for having hope for the future, to overcome the limitation of the previous study conducted in Japan.

Further, we found that hope for the future was significantly associated with time spent on reading books and studying except in class. That is, these results indicate that students who have hope for the future might be able to manage their time more effectively to achieve the desired outcome. It is also possible that students who have already decided what they want to do in the future tend to consider what they have to do now and manage their time accordingly. Our findings are consistent with previous ones about future time perspective [11], which is a similar concept to hope for the future. In the previous study, it was reported that future positiveness and future plans, which are components of future time perspective, have significant positive correlations with time management disposition. Another important point of this previous study is that hope for the future can be inspired by adults in a comparatively easy way.

In addition, it was previously shown that time planning has a significant indirect effect on academic achievement through time efficacy [11]. Considering this result, the association between having hope for the future and academic performance might be mediated by reading books and studying except in class. A further randomized controlled study of the effectiveness of hope for the future and its association with time management is warranted.

The current study also revealed that hope for the future has a positive association with resilience. This result is also consistent with the previous study, which revealed the association between future time perspective and resilience [42]. The students who have hope for the future might be able to imagine their future vision or future goals more clearly, and their will to realize their dreams might enable them to develop their patience. Further studies using hope for the future as an intervention, especially among those who need to develop resilience, such as those experiencing childhood adversity [43], are warranted.

Several limitations need to be addressed. First, the details of individual hope for the future were not investigated. Hope for the future was determined by the response to one question only. In addition, academic performance was subjectively measured. Future studies need to objectively assess academic performance as well as the types of jobs or social status the child hopes to have in the future. Second, the current study cannot determine the causal association because reverse causation is likely to have occurred due to the cross-sectional nature of the study, even though propensity score matching analysis was used. Those who have better academic performance may be more likely to have hope for the future. Furthermore, unmeasured confounders, such as child temperament, genetic factors, or childhood environment, were not addressed. We need a further longitudinal study to elucidate the mechanisms of whether hope for the future promotes better future academic performance.

## 5. Conclusions

This study highlighted that having hope for the future was associated with better academic performance among adolescents. This association may be partially explained by reading books, studying except in class, and resilience. A further study investigating the details of individual hope for the future or objective academic performance is warranted to show a more precise inference on the association. Nonetheless, the current findings provide important information on encouraging adolescents to have hope for the future to promote better academic performance.

## Figures and Tables

**Figure 1 ijerph-19-11890-f001:**
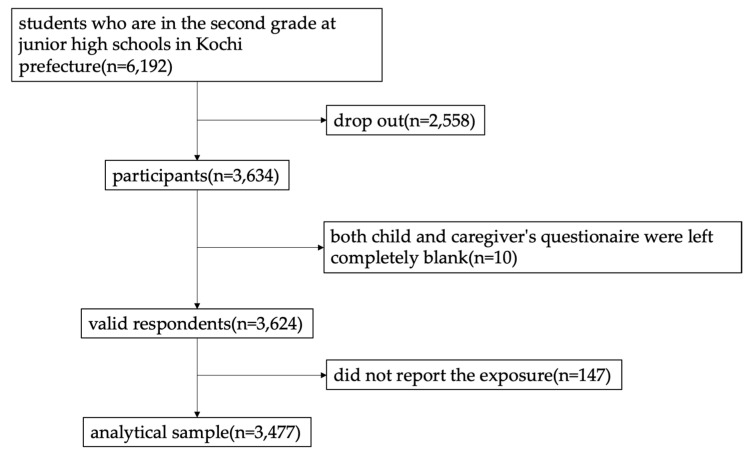
Requirement flow chart.

**Table 1 ijerph-19-11890-t001:** Characteristics of covariates before and after propensity score matching.

		Before PS Matching						After PS Matching					
		Hope for the Future						Hope for Future					
		Yes (*n* = 2283; 65.7%)		No (*n* = 1194; 34.3%)		*p* Value	Bias (%)	Yes (*n* = 1105)		No (*n* = 1105)		*p* Value	Bias (%)
		*n*	%	*n*	%			*n*	%	*n*	%		
Child’s sex						0.003						0.625	
	boy	1009	44.2	592	49.6			557	50.4	538	48.7		
	girl	1220	53.4	566	47.4		12.1	521	47.2	535	48.4		−2.5
	Missing	54	2.4	36	3.0		−4.0	27	2.4	32	2.9		−2.8
Child’s height						0.137						0.882	
	<150	216	9.5	90	7.5			72	6.5	83	7.5		
	150–<160	935	41.0	472	39.5		2.9	436	39.5	440	39.8		−0.7
	160–<170	752	32.9	422	35.3		−5.1	409	37.0	394	35.7		2.9
	170+	219	9.6	110	9.2		1.3	99	9.0	102	9.2		−0.9
	Missing	161	7.1	100	8.4		−5.0	89	8.1	86	7.8		1.0
Child’s weight						0.267						0.996	
	<40	178	7.8	100	8.4			86	7.8	91	8.2		
	40–<50	984	43.1	505	42.3		1.6	471	42.6	473	42.8		−0.4
	50–<60	617	27.0	340	28.5		−3.2	320	29.0	316	28.6		0.8
	60+	252	11.0	106	8.9		7.2	100	9.1	99	9.0		0.3
	Missing	252	11.0	143	12.0		−2.9	128	11.6	126	11.4		0.6
Maternal age						0.518						0.918	
	<40	379	16.6	188	15.8			175	15.8	175	15.8		
	40–<50	1526	66.8	785	65.8		2.3	735	66.5	733	66.3		0.4
	50+	229	10.0	132	11.1		−3.3	112	10.1	120	10.9		−2.4
	Missing	149	6.5	89	7.5		−3.6	83	7.5	77	7.0		2.1
Paternal age						0.239						0.994	
	<40	215	9.4	109	9.1			101	9.1	99	9.0		
	40–<50	1224	53.6	640	53.6		0	600	54.3	596	53.9		0.7
	50+	405	17.7	240	20.1		−6.0	214	19.4	218	19.7		−0.9
	Missing	439	19.2	205	17.2		5.3	190	17.2	192	17.4		−0.5
Maternal education						0.385						0.943	
	High school or less	826	36.2	402	33.7			375	33.9	372	33.7		
	Some college	972	42.6	524	43.9		−2.6	475	43.0	490	44.3		−2.7
	College or more	335	14.7	178	14.9		−0.7	165	14.9	162	14.7		0.8
	Other/Unknown	4	0.2	5	0.4		−4.5	3	0.3	3	0.3		0
	Missing	146	6.4	85	7.1		−2.9	87	7.9	78	7.1		3.2
Paternal education						0.300						0.904	
	High school or less	921	40.3	486	40.7			425	38.5	441	39.9		
	Some college	360	15.8	197	16.5		−2.0	197	17.8	183	16.6		3.4
	College or more	559	24.5	311	26.1		−3.6	288	26.1	288	26.1		0
	Other/Unknown	12	0.5	3	0.3		4.4	2	0.2	3	0.3		−1.5
	Missing	431	18.9	197	16.5		6.2	193	17.5	190	17.2		0.7
Maternal job						0.259						0.951	
	Full-time	837	36.7	418	35.0			401	35.6	387	34.4		
	Part-time	805	35.3	416	34.8		0.9	396	35.2	404	35.9		−2.3
	Others	266	11.7	128	10.7		3.0	118	10.5	115	10.2		2.0
	Not working	243	10.6	150	12.6		−6.0	138	12.3	140	12.4		0.8
	Missing	132	5.8	82	6.9		−4.5	73	6.5	80	7.1		3.7
Paternal job						0.065						0.998	
	Full-time	1395	61.1	730	61.1			700	62.2	702	62.3		
	Part-time	74	3.2	43	3.6		−2.0	40	3.6	37	3.3		−1.0
	Others	358	15.7	207	17.3		−4.5	186	16.5	186	16.5		2.7
	Not working	24	1.1	22	1.8		−6.6	14	1.2	14	1.2		−3.0
	Missing	432	18.9	192	16.1		7.5	186	16.5	187	16.6		0.7
Marital states						0.089						0.941	
	Married	1829	80.1	988	82.8			905	81.9	910	82.4		
	Unmarried	429	18.8	199	16.7		5.6	192	17.4	188	17.0		0.9
	Missing	25	1.1	7	0.6		5.6	8	0.7	7	0.6		1.0
Annual household income (JPY)						0.340						0.926	
	<3 million	448	19.6	209	17.5			179	16.2	190	17.2		
	3 million–<6 million	701	30.7	364	30.5		0.5	338	30.6	333	30.1		1.0
	6 million+	716	31.4	385	32.2		−1.9	361	32.7	362	32.8		−0.2
	Unknown	99	4.3	46	3.9		2.4	40	3.6	44	4.0		−1.8
	Missing	319	14.0	190	15.9		−5.4	187	16.9	176	15.9		2.8
Public assistance						0.207						0.926	
	No	1388	60.8	761	63.7			712	63.2	717	63.7		
	Yes	16	0.7	6	0.5		2.6	5	0.4	6	0.5		−1.2
	Missing	879	38.5	427	35.8		5.7	409	36.3	403	35.8		−2.1
Economic difficulties						0.387						0.843	
	No	1908	83.6	990	82.9			928	84.0	918	83.1		
	Yes	286	12.5	165	13.8		−3.8	144	13.0	153	13.9		−2.4
	Missing	89	3.9	39	3.3		3.4	33	3.0	34	3.1		−0.5
Lack of daily necessities for children						0.214						0.979	
	No	1341	58.7	698	58.5			655	59.3	651	58.9		
	Yes	515	22.6	295	24.7		−5.1	260	23.5	273	23.9		−0.9
	Missing	427	18.7	201	16.8		4.9	190	17.2	187	17.2		0.0
Lack of daily necessities						0.313						0.925	
	No	1419	62.2	760	63.7			707	64.0	702	63.5		
	Yes	396	17.4	215	18.0		−1.7	187	16.9	194	17.6		−1.7
	Missing	468	20.5	219	18.3		5.5	211	19.1	209	18.9		0.5
Caregiver’s physical health						0.120						0.947	
	Good	1299	56.9	693	58.0			637	57.7	644	58.3		
	Average	590	25.8	315	26.4		−1.2	299	27.1	288	26.1		2.3
	Poor	218	9.6	120	10.1		−1.7	105	9.5	110	10.0		−1.5
	Missing	176	7.7	66	5.5		8.8	64	5.8	63	5.7		0.4
Caregiver’s mental health (K6)						0.506						0.935	
	<5	1538	67.4	834	69.9			776	70.2	767	69.4		
	5–<13	548	24.0	265	265.0		4.3	242	21.9	251	22.7		−1.9
	13+	116	5.1	54	54.0		2.6	46	4.2	49	4.4		−1.3
	Missing	81	3.6	41	41.0		0.6	41	3.7	38	3.4		1.5
Maternal smoking habit						0.476						0.951	
	Smoking now	337	14.8	155	13.0			155	14.0	152	13.8		
	Smoking past	332	14.5	187	15.7		−3.1	161	14.6	180	15.4		−2.3
	No history	1470	64.4	774	64.8		−0.9	720	65.2	718	64.4		1.5
	Missing	144	6.3	78	6.5		−0.9	69	6.2	71	6.4		−0.7
Paternal smoking habit						0.103						0.960	
	Smoking now	742	32.5	385	32.2			353	32.0	363	32.9		
	Smoking past	613	26.9	361	30.2		−7.5	330	29.9	320	29.0		2.0
	No history	505	22.1	257	21.5		1.4	239	21.6	240	21.7		−0.2
	Missing	423	18.5	191	16.0		6.7	183	16.6	182	16.5		0.2
Living with maternal grandmother						0.112						0.686	
	No	2085	91.3	1109	92.9			1020	92.3	1025	92.8		
	Yes	198	8.7	85	7.1		5.8	85	7.7	80	7.2		1.7
Living with maternal grandfather						0.131						0.607	
	No	2150	94.2	1139	95.4			1058	95.8	1053	95.3		
	Yes	133	5.8	55	4.6		5.5	47	4.3	52	4.7		−2.0
Living with paternal grandmother						0.095						0.606	
	No	2029	88.9	1083	90.7			1007	91.1	1000	90.5		
	Yes	254	11.1	111	9.3		6.0	98	8.9	105	9.5		−2.1
Living with paternal grandfather						0.535						0.931	
	No	2127	93.2	1119	93.7			1034	93.6	1035	93.7		
	Yes	156	6.8	75	6.3		2.2	71	6.4	70	6.3		0.4
Older brother						0.242						0.847	
	No	1709	74.9	872	73.0			810	73.3	814	73.7		
	Yes	574	25.1	322	27.0		−4.2	295	26.7	291	26.3		0.8
Older sister						0.102						0.731	
	No	1764	77.3	893	74.8			826	74.8	833	75.4		
	Yes	519	22.7	301	25.2		−5.8	279	25.3	272	24.6		1.5
Younger brother						0.103						0.962	
	No	1615	70.7	876	73.4			807	73.0	806	72.9		
	Yes	668	29.3	318	26.6		5.9	298	27.0	299	27.1		−0.2
Younger sister						0.758						0.962	
	No	1667	73.0	866	72.5			801	72.5	800	72.4		
	Yes	616	27.0	328	27.5		−1.1	304	27.5	305	27.6		−0.2
Child’s depression (DSRS)						<0.001						0.868	
	No (0–15)	2018	88.4	973	81.5			919	83.2	928	84.0		
	Yes (16+)	230	10.1	209	17.5		−21.7	176	15.9	168	15.2		2.1
	Missing	35	1.5	12	1.0		4.7	10	0.9	9	0.8		0.8

**Table 2 ijerph-19-11890-t002:** Distribution of answers about time usage before and after propensity score matching.

			Before PS Matching	After PS Matching
	Hope for the Future		*n* (%)	*n* (%)
Time of playing games	Yes	Less than 1 h/day (=0)	1602 (70.2)	767 (69.4)
		More than 1 h/day (=1)	619 (27.1)	315 (28.5)
		Missing	62 (2.7)	23 (2.1)
	No	Less than 1 h/day (=0)	766 (64.1)	720 (65.2)
		More than 1 h/day (=1)	396 (33.2)	357 (32.3)
		Missing	32 (2.7)	28 (2.5)
Time to watch TV or DVD	Yes	Less than 1 h/day (=0)	679 (29.7)	334 (30.2)
		More than 1 h/day (=1)	1538 (67.4)	742 (67.1)
		Missing	66 (2.9)	29 (2.6)
	No	Less than 1 h/day (=0)	382 (32.0)	352 (31.9)
		More than 1 h/day (=1)	774 (64.8)	719 (65.1)
		Missing	38 (3.2)	34 (3.1)
Reading time	Yes	Almost never (=0)	545 (23.9)	266 (24.1)
		More than 1day/ week (=1)	1660 (72.7)	807 (73.0)
		Missing	78 (3.42)	32 (2.9)
	No	Almost never (=0)	380 (31.8)	348 (31.5)
		More than 1day/week (=1)	772 (64.7)	719 (65.1)
		Missing	42 (3.5)	38 (3.4)
Studying time except in class	Yes	Almost never (=0)	146 (6.4)	65 (5.9)
		More than 1day/week (=1)	2089 (61.5)	1034 (93.6)
		Missing	48 (2.1)	6 (0.5)
	No	Almost never (=0)	128 (10.7)	117 (10.6)
		More than 1day/week (=1)	1045 (87.5)	981 (88.8)
		Missing	21 (1.8)	7 (0.6)

**Table 3 ijerph-19-11890-t003:** Statistical results of academic performance and resilience before and after propensity score matching.

		Before PS Matching							After PS Matching			
			Crude Model			Adjusted Model						
		Mean (SD)	β	95%CI	*p* Value	β	95%CI	*p* Value	Mean (SD)	β	95%CI	*p* Value
Self-rated academic performance in class	Yes	3.03 (1.27)	0.21	0.12 to 0.30	<0.001	0.19	0.10 to 0.28	<0.001	3.02 (1.28)	0.21	0.10 to 0.32	<0.001
	No	2.82 (1.28)							2.81 (1.28)			
Child’s resilience (CRCS total score)	Yes	18.8 (5.58)	1.77	1.35 to 2.19	<0.001	1.51	1.11 to 1.91	<0.001	19.1 (5.82)	1.48	0.98 to 1.98	<0.001
	No	20.6 (5.82)							20.8 (6.01)			

**Table 4 ijerph-19-11890-t004:** Statistical results of time usage before and after propensity score matching.

	Before PS Matching						After PS Matching		
	Crude Model			Adjusted Model					
	OR	95%CI	*p* Value	OR	95%CI	*p* Value	OR	95%CI	*p* Value
Time of playing games	0.75	0.64 to 0.87	<0.001	0.81	0.69 to 0.97	0.018	0.81	0.67 to 0.98	0.028
Time to watch TV or DVDs	1.12	0.96 to 1.30	0.15	1.06	0.91 to 1.25	0.449	1.05	0.87 to 1.26	0.61
Reading books	1.50	1.28 to 1.75	<0.001	1.50	1.27 to 1.77	<0.001	1.45	1.19 to 1.75	<0.001
Studying except in class	11.7	7.93 to 15.5	<0.001	1.65	1.26 to 2.14	<0.001	1.89	1.37 to 2.61	<0.001

## Data Availability

Not applicable.

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
