# Peer review of "Association between Hope for the Future and Academic Performance in Adolescents: Results from the K-CHILD Study"

_ijerph, 2022, doi:10.3390/ijerph191911890_

Round 1
Reviewer 1 Report
Literature Review is missing. Literature is necessary to explain how this aim came about - An additional aim of this study was to examine 55 the effects of adolescents’ time usage (i.e., studying, reading and screen time) and their 56 resilience, a mediating factor for the association between hope for the future and academic 57 performance.
The following should be in the Literature Review section-
The 197 association between hope for the future and academic performance has not been revealed 198 among previous studies. For example, in China and Iran, only the significant association 199
between future orientation or planning and academic achievement was demonstrated [16, 200
17].
Our findings are consistent with previous ones about future time 207 perspective, which is a similar concept to hope for the future (18). In that study, the au- 208 thors reported that future positiveness and future plans, which are components of future 209 time perspective, have significant positive correlations with time management disposi- 210 tion. It is also possible that students who have already decided what they want to do in 211 the future tend to consider what they have to do now, and to manage their time accord- 212 ingly. However, it was previously shown that time planning has a significant indirect ef- 213fect on academic achievement through time efficacy [18].
The current study also revealed that hope for the future job has favorable association 219
with resilience. This result is also consistent with the previous study, which revealed the 220
association between future time perspective and resilience [19]. The students who have 221
hope for the future job might be able to image their future vision or future goal more 222
clearly, and their will to realize their dreams might enable them to develop their patience. 2
Author Response
Thank you for your valuable comments. As you suggested, we revised our manuscript. Please see the attached file.

Reviewer 2 Report
Overall, I found the presented manuscript to be well developed and to have the potential to make a meaningful contribution to the literature. There are, however, a series of theoretical and methodological issues that need to be considered.
1. There is hardly any theoretical background preparing the investigated research questions.
2. In particular, the authors conducted hypotheses-testing analyses, yet they do not include the hypotheses in the manuscript.
3. In particular, all covariates need to be carefully prepared theoretically, in order to justify their inclusion.
4. The authors transform cateogorical data into interval scaled data to be able to conduct the analyses. This needs to be justified and is potentially problematic. Given the large dataset, also other estimation procedures could be used, retaining the original data format. This would be preferrable.
5. It does not become clear why Propensity Score Matching was used here. For the conducted analyses, the covariates could simply be included as covariates.
6. The p values need to be adjusted for the multiple individual analyses that are conducted, or a multivariate analyses should be conducted.
7. With hope for the future being a dichotomous variable, it is not clear how this was handeled in the first set of analyses.
8. I suggest including standardized regression weights in Table 2 for better comparability of the yielded effects.
9. Like the theoretical background being largely missing, an actual discussion of the results and how they contribute to prior research on this topic, and out theoretical reasoning is largely lacking.
10. Extensive editing of the language is necessary. I had issues understanding many of the sentences in the manuscript.
Author Response
Thank you for your helpful comments. Please see the attached file.

Reviewer 3 Report
The manuscript addresses a very interesting international topic that can help to improve academic performance, under the title "Association between hope for the future and academic performance in adolescents: Results from the K-CHILD study". The topic of the manuscript, as well as the updated literature review provided by the authors, is appropriate.
The overall assessment of the paper is positive. The authors have pointed out the importance of hope for the future on the academic performance of Japanese adolescents. However, I suggest some points that could be improved.
It is noted that the article is unbalanced. Especially in the results section.
INTRODUCTION. The authors address the topic of the study in a clear manner and rely on a review of the scientific literature that is neither comprehensive nor current. This section could be significantly improved by adding high impact scientific articles.
METHOD. The research design is correct and the data used seem useful for the objectives set out by the author of the text.
RESULTS. I consider that this section needs a major revision.
The statistical tools used are simple and the analysis of the results is scarce and shallow. There are also many tables and little explanation of the results.
CONCLUSION
This section needs a minor revision. I suggest moving the limitations to the conclusion section.
Overall, it is a good piece of work and I hope my suggestions will help.
Author Response
Thank you for your informative suggestions. As you suggested, we revised our manuscript. Please see the attached file.

Round 2
Reviewer 1 Report
You have addressed the concerns I raised.
Author Response
We appreciate your helpful comments.
Reviewer 2 Report
This is my second review of this manuscript. Regarding the first version, I pointed to a series of theoretical and methodological issues that required attention. The authors have revised most of them, improving the overall quality of the manuscript. Referring to my original points:
3. A clear argument needs to be made with regard to each covariate why this should be related to both the predictor and the outcome. Failing to do so does not warrant inclusion of such variables as covariates. Further, robustness checks for the inclusion of these variables should be added.
4. Ease of interpretability is not a good reason for changing the statistical premises. An argument would need to be made why the analyses can be trusted despite data actually existing in an ordinal-scaled and not an interval-scaled data format. I suggest running the analyses with another type of estimator that can handle ordinal data.
6. My comment was not about adding p values but about adjusting them for the multiple separate tests.
8. The "standardized beta values" are much partly much higher or much lower than 1 and -1 respectively.
9. Both the theoretical background and the discussion is still rather weak. Besides situating the presented findings against other research, it should be elaborated on what the presented findings imply for theory and future research.
